# A supervised multiclass framework for mineral classification of Iberian beads

**Daniel Sanchez-Gomez**[1]*, **Carlos P. Odriozola Lloret**[1,2], **Ana Catarina Sousa**[1], **José Ángel Garrido-Cordero**[2], **Galo Romero-García**[2], **José María Martínez-Blanes**[3,4], **Manel Edo I. Benaiges**[5], **Rodrigo Villalobos-García**[6], **Victor S. Gonçalves**[1]

**1** Centro de Arqueologia da Universidade de Lisboa (UNIARQ), Lisbon, Portugal, **2** Dpto. de Prehistoria y Arqueología, Universidad de Sevilla, Seville, Spain, **3** Instituto de Ciencia de Materiales de Sevilla, Universidad de Sevilla- Consejo Superior de Investigaciones Científicas, Seville, Spain, **4** Dpto. de Química Inorgánica, Universidad de Sevilla, Seville, Spain, **5** Institut d'Arqueologia, Universitat de Barcelona, Barcelona, Spain, **6** Cuerpo de Profesores de Enseñanza Secundaria, Gobierno de Cantabria, Cantabria, Spain

\* daniel-sanchez-gomez@edu.ulisboa.pt

**Data Availability Statement:** All source code is available at https://github.com/Daniel-SanchezG/MACLAS (DOI 10.5281/zenodo.10155331) A

## Abstract

Research on personal adornments depends on the reliable characterisation of materials to trace provenance and model complex social networks. However, many analytical techniques require the transfer of materials from the museum to the laboratory, involving high insurance costs and limiting the number of items that can be analysed, making the process of empirical data collection a complicated, expensive and time-consuming routine. In this study, we compiled the largest geochemical dataset of Iberian personal adornments ($n =$ 1243 samples) by coupling X-ray fluorescence compositional data with their respective X-ray diffraction mineral labels. This allowed us to develop a machine learning-based framework for the prediction of bead-forming minerals by training and benchmarking 13 of the most widely used supervised algorithms. As a proof of concept, we developed a multiclass model and evaluated its performance on two assemblages from different Portuguese sites with current mineralogical characterisation: Cova das Lapas ($n = 15$ samples) and Gruta da Marmota ($n = 10$ samples). Our results showed that decisión-tres based classifiers outperformed other classification logics given the discriminative importance of some chemical elements in determining the mineral phase, which fits particularly well with the decision-making process of this type of model. The comparison of results between the different validation sets and the proof-of-concept has highlighted the risk of using synthetic data to handle imbalance and the main limitation of the framework: its restrictive class system. We conclude that the presented approach can successfully assist in the mineral classification workflow when specific analyses are not available, saving time and allowing a transparent and straightforward assessment of model predictions. Furthermore, we propose a workflow for the interpretation of predictions using the model outputs as compound responses enabling an uncertainty reduction approach currently used by our team. The Python-based framework is packaged in a public repository and includes all the necessary resources for its reusability without the need for any installation.

extended version of the dataset is within
Supporting information Files.

**Funding:** This work has been financed with
Portuguese funds through FCT - Fundação para a
Ciência e a Tecnologia in the framework of the
projects UIDB/00698/2020 (https://doi.org/10.
54499/UIDB/00698/2020) and UIDP/00698/2020
(https://doi.org/10.54499/UIDP/00698/2020).
(DSG) has received resources from the Fundação
para a Ciência e Tecnologia de Portugal (www.fct.
pt) in the framework of the doctoral grant (UI/BD/
154365/2023). The Spanish Ministry of Science
and Technology has funded this study through the
research project (PID2021-124421NB-I00) (https://
investigacion.us.es/sisius/sis_proyecto.php?
idproy=33567) whose PI is (CPOLI) and research
team is composed by (JMMB), (GRG), and (JAGC).
The other authors do not declare having received
funding for their participation in this project. The
funders had no role in study design, data collection
and analysis, decision to publish, or preparation of
the manuscript.

**Competing interests:** The authors declare that they
have no known competing financial interests or
personal relationships that could have appeared to
influence the work reported in this paper

## Introduction

The study of personal adornments (beads, pendants, or charms) has proven important for understanding the emergence of social complexity in the past [1–4]. In this sense, reliable identification of materials is essential for obtaining information on material flows between sources and destinations, modelling provenance and complex social networks as well as for a better understanding of the technological knowledge and symbolic signalling related to their consumption.

Since the 19th century, scholars have focused on callaite or "perles du callaïs", one of the raw materials used as a marker of prestige and long-distance exchange in Western European prehistory to model past social interaction [5, 6]. However, from the beginning, scholars faced a problem by equating callaite, which is actually a generic term first used by Pliny in his Historia Naturalis to describe any type of green beads (micas, talcs, chlorites, among others), with variscite, a special type of phosphate whose rare sources and particular chemistry make it a perfect proxy for provenance studies and exchange network analysis. It was only until some pioneering studies made the first X-ray diffraction analyses that it became clear that not all green beads were made of variscite and not all variscite beads are green [7–9].

The lack of empirical data both on the composition of the raw materials and on the exact number of pieces that compose the assemblages remains, to date, a critical bottleneck in the aim of disentangling the complex social dynamics surrounding the consumption of these objects in the prehistory of much of western Europe, particularly between the Early Neolithic and Late Bronze Age, when a striking increase in the consumption of mineral ornaments (among others materials) is evidenced by the thousands of beads found in sites of very diverse typology [10–12].

The absence of specialised analysis is often replaced by a coarse-grained visual classification (e.g. shale/slate, limestone or sandstone) which is useful for general descriptions, but severely limits the possibilities to move towards a deeper understanding of the diversity of these artefacts and their consumption dynamics. In the case of callaite, this has left the ambiguous and unreliable category of "green stones" as the only possibility to cluster a heterogeneous set of materials of great social and economic relevance, which obscures the representativeness and reliability of current network analyses and provenance models [10–13].

Furthermore, although long disregarded, beads of several compositions were extensively used during prehistory and craftsmen's rock choices are shown to be much more diverse than green hues, accounting for rocks of many colours, drawing a much more complex picture than expected and and pointing out the need for a better characterisation of this type of proxy in order to understand the causes behind the selection of certain raw materials over others, the uneven distribution of raw materials among different types of sites, as well as to elucidate whether certain raw materials can be used as chrono-cultural markers to construct more specific time series for the circulation of these artefacts.

In recent decades, the study of personal adornments in the Iberian Peninsula has explored different approaches that have opened the way to a fertile multidisciplinary research path that addresses mineral classification regardless of colour [14–17].

Among these methodological paths, spectroscopic techniques have become essential because they allow precise identification of the elemental and molecular characteristics of the artefacts. Some of the most widely used techniques for bead characterisation include X-ray fluorescence (XRF), X-ray diffraction (XRD), Raman spectroscopy, Proton-induced X-ray emission (PIXE), Fourier-Transform Infrared Spectroscopy (FTIR), Near Infrared Spectroscopy (NIR), Nuclear Magnetic Resonance (NMR). But there is one that stands out because of its versatility in the analysis of different types of samples, its portability, and low cost: portable

X-ray fluorescence (p-XRF). For these reasons, its popularity among archaeologists has grown considerably and today it is common to find this type of device in archaeological laboratories worldwide. As it allows the elemental composition of the analyzed object to be easily and non-destructively determined in situ, it is often used as a first approach to material characterisation analysis. However, as handy as it may be, pXRF cannot determine the molecular composition of the object and therefore is unable to determine the mineralogy of the beads. As this is usually an important part of any archaeometric study, elemental characterization is frequently complemented by other mineralogic techniques, such as XRD, FTIR, NIR, or Raman. Nevertheless, the drawback of most of these powerful methods is that they are expensive, usually not portable and the interpretation of the results is time-consuming and not straightforward. Furthermore, requests for the loan of archaeological materials to museums are slow, complicated and generally unsuccessful.

This has motivated us to explore alternative approaches to address the issues associated with the fast and reliable identification of beads and data collection.

In recent years we have created a database that to date has (n = 1243) compositional and mineralogical analyses by pXRF and XRD of elements of personal adornment from the Iberian Peninsula. This has allowed us to investigate a supervised approach for mineral classification based on the most widely used technology currently available, pXRF, thus avoiding the loan of artefacts from museums, the risk associated with transporting them to laboratories, and the added costs of using large infrastructures and long analysis times.

Machine learning have become a powerful alternative in archaeology thanks to the increasing computational power, the accessibility of algorithms and the growing amount of data that is constantly emerging [18–20].

Integration of archaeometric techniques with different ML methods has become a fertile avenue for data analysis and predictive modelling in recent years. Workflows that implement supervised methods in combination with different analytical techniques have been successfully applied to the classification and provenance of archaeological ceramics [21–23], obsidians [24], soils [25], or geological variscites [9]. However, the use of ML for the study of personal adornments has not yet been sufficiently explored, even though archaeometric and geochemical data are continuously increasing.

In this paper, we present a supervised multiclass framework for the mineral classification of prehistoric beads that can be used to infer mineral species from a matrix of compositional data. We trained and compared some of the most commonly used algorithms for classification tasks in a workflow that is reproducible and reusable without the need to install any piece of software. As a proof-of-concept, we chose the best performer to carry out a real use case with two sample sets from different sites.

## Materials and methods

The applied methods of this study encompass p-XRF and XRD sample measurements, data processing and algorithm experimentation.

### The data

The dataset used in this work has been constructed by coupling the elemental and mineral information of ($n$ = 1243 samples) from 52 sites in Portugal and Spain in a chronological span from 5th to 3rd millennia BC (Fig 1) (S1 Dataset). Each data point is described by a vector of 45 independent variables with the chemical elemental concentrations expressed in atomic percentages obtained by p-XRF associated with a single categorical mineral label obtained by XRD as the dependent variable (S1 Appendix).

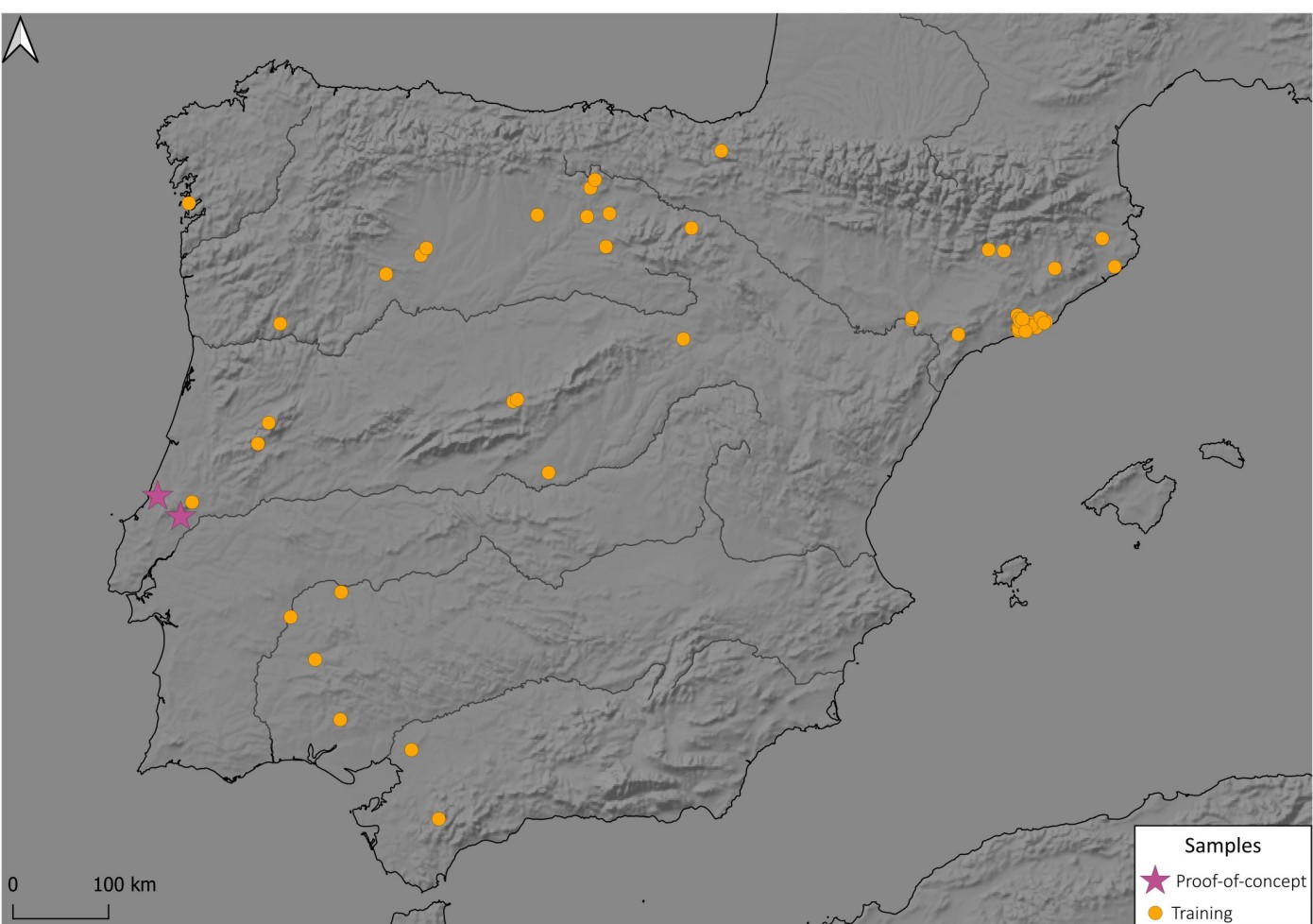

**Fig 1. Locations of archaeological sites included in the dataset: Due to geographical coordinates overlapping, not all archaeological locations are visible.** Maps were made using the software QGIS [26] 3.34 and Natural Earth raster maps [27].

The data was collected at different academic centres, museums, and reference collections in Spain and Portugal, as well as from objects from our surveys and excavations [12, 16, 28, 29]. It should be noted that there is a bias towards the over-representation of variscites ($\sim 63\%$) (n = 784) in the data given the scientific interest that this type of phosphate has aroused. (Table 1).

**XRF measurements.** Chemical composition was measured by focusing directly on the sample on an area of $100mm^2$ with an Oxford Instruments XMET-7500 pXRF equipped with an Rh tube, a silicon drift detector (SDD), and an automatic 5-position filter changer. Quantification was obtained using the SOILS-LE program based on the fundamental parameter (FP) method. The calibration of the measurements at the beginning and the end of each run was done by measuring 32-mm pellets of a European Commission Institute for Reference Materials and Measurements Natural Moroccan phosphate standard (BCR-032 #919). Data and statistics on reference materials can be found in the (S1 Dataset) and in the supplementary material (S1 Appendix).

**XRD analysis.** Mineral compositions were determined using a Panalytical X'Pert Pro θ/θ diffractometer equipped with Cu Kα source (1.5406 Å) operating at 45 kV and 40 mA. A

**Table 1. Class system distribution.**

| Principal Classes | Mineral species | No of samples |
|---|---|---|
| Phosphates | Aheylite | 34 |
| | Berlinite | 26 |
| | Variscite | 784 |
| | Crandallite | 25 |
| | Planerite | 18 |
| | Turquoise | 7 |
| | Strengite | 12 |
| | Annabergite | 11 |
| | Metavariscite | 7 |
| Silicates | Talc | 41 |
| | Muscovite | 73 |
| | Chlorite-serpentine | 35 |
| | Illite | 7 |
| | Clinochlore | 52 |
| Halides | Fluorite | 8 |
| Carbonates | Calcite | 27 |
| | Aragonite | 8 |
| Oxides | Quartz | 8 |

The phosphate group is over-represented because much of the research on mineral beads in the Iberian Peninsula has revolved around variscite as a proxy for provenance studies. We tried to individualise as many classes as possible, including some polymorphs such as Metavariscite to add complexity to the model and try to achieve greater granularity in the predictions. In the case of quartz, although it frequently occurs together with many other mineral phases, some translucent beads are made mainly of this material, and therefore its inclusion in the system was justified.

PixCel detector was used and the data were collected in transmission mode with a 2D detector. An incident beam PreFIX module with an X-ray mirror for Cu radiation was used to allow non-destructive analysis.

### Proof-of-concept

In order to test the model in a real-world case, ($n = 25$) samples of two different Portuguese sites were used: Cova das Lapas ($n = 15$ samples) and Gruta da Marmota ($n = 10$ samples).

The samples are part of the materials recovered from the excavation campaigns carried out by the team members (ACS and VSG), thus, no loan permits were required.

Chemical and mineralogical data was obtained following the same procedure described for XRF measurements and XRD analysis and is provided within the dataset. In this way we were able to compare the results obtained by the described analytical techniques that constitute the ground truth (S1 File) with the predictions of the model, which allowed us to assess the real generalisation capacity of the model to unseen data.

### Data processing

Before any analysis, all data were pre-processed according to the following transformations:

- All elements were put into atomic percentages per chemical element;

- Below detection limit values (BDL) were replaced with an informed guess (LoD/$\sqrt{2}$ [30]) since most ML methods do not work with missing values [31]. To calculate the imputation values, we used the device´s limits of detection (LoD) for a SiO2 matrix and estimated their conversion from parts per million (ppm) to atomic percent to fit the model´s input format (S1 Dataset). Although these are near-zero values negligible from a statistical standpoint, we wanted to replace the missing values with a constant that would produce the smallest bias in the distribution according to [30]

- The composition of each sample was closed to 100%.

- Elements from Mg to Th were kept in the pipeline

**Resampling.**   A compound strategy of class deletion and a resampling technique was implemented to reduce the coefficient of variation in the data [32–35]. This was required because the overrepresentation of variscites generated a class imbalance of around two orders of magnitude in the training set. To address this situation, we first set a threshold of seven cases per class to keep the class in the model. Below this threshold, the variability of the data within the classes is severely affected for resampling purposes. Therefore, the cardinality of the dependent variable contains ($n$ = 18) mineral classes that constitute the class system of the model. Then, a hybrid technique of undersampling the majority class and oversampling the minority classes was applied. When undersampling, the condensed nearest neighbour rule [36] was used to retain only non-redundant data points and avoid information loss and, when oversampling, a synthetic minority oversampling technique (SMOTE) was implemented to avoid loss of variability [33]. As a rule of thumb, an odd number of 5 neighbours was used in both cases.

## Model development and class prediction

Manual labelling of the training set was based on the criterion of choosing the main phase obtained in the XRD diffractogram when more than one mineral was recorded, as most rock types are composed of more than one mineral and contain secondary crystallographic phases and amorphous materials (e.g., cases where labelled samples appeared as Variscite + Fe oxide, Variscite + Quartz, or Calcite + Fe oxide were labelled as Variscites or Calcites respectively).

After data preparation, the final shape of the dataset used for model development was 4082 data points with 45 independent variables and two possible targets; major mineral groups ($n$ = 5 classes) and mineral species ($n$ = 18 classes).

Thirteen different algorithms were trained and compared (Table 2). These models were chosen among the most widely used for supervised classification tasks when using tabular data. A standard procedure of splitting was performed using 80% for training and 20% for testing the model. A $k$-fold stratified cross-validation strategy was used for evaluation with standard ($k$ = 10) iterations [37]. Although its effectiveness in estimating model performance makes cross-validation a standard technique for assessing ML models, it has been noted that constant exposure to the same data for training, testing and tuning can lead to overestimating model performance [38]. For this reason, two different validation sets VS1 ($n$ = 118) and VS2 ($n$ = 453) were created as final validation filters within the pipeline. Each contains ten per cent of the data before and after resampling respectively. These data were not considered in the training process and a small sample of the original data (VS1) without any preprocess was available as a final filter (Fig 2).

The performance of all models was evaluated by comparing some of the most widely used metrics (Precision, Recall and F1-score) in addition to their accuracy since it is well known

**Table 2. Algorithms used in this study.**

| Type of Model | Algorithm |
| --- | --- |
| Decision Tree | CART [39] |
| Bagging Ensemble Models | Random Forest Classifier [40] |
| | Extra Trees Classifier [41] |
| Boosting Ensemble Models | LightGradientBoostingMachine [42] |
| | Extreme Gradient Boosting [43] |
| | Ada Boost Classifier [44]https://www.zotero.org/google-docs/?SipXl0 |
| Discriminant Analysis | Linear discriminant analysis [43, 45, 46] |
| | Quadratic discriminant analysis [43, 45, 46] |
| K Neighbors | K Neighbors Classifier [45] |
| Ridge Regression | RidgeClassifier [43] |
| Naive Bayes | GaussianNB [43] |
| Logistic Regression | LogisticRegression [43] |

that for classification problems with imbalanced datasets, overall accuracy does not reliably reflect the model's performance.

The best model was selected through the same resampling technique (stratified cross-validation) for further procedures which consisted of hyperparameter optimisation using a randomized search algorithm, probability calibration using a sigmoid regressor to obtain reliable probabilistic predictions and a last iteration of training on the entire dataset (training + test) before its deployment [43, 44, 46].

## Data and code availability

No human remains were used in the present study. No permits were required for the analysis of geological samples. Permits for the analyses of archaeological specimens were granted by the Chief Curators and Heads of the museums listed as supporting information in the dataset description (S1 Appendix) A repository with our source code and training data, conveniently packaged with README instructions and links for reuse, is publicly available on the GitHub page (https://github.com/Daniel-SanchezG/MACLAS). The repository includes different

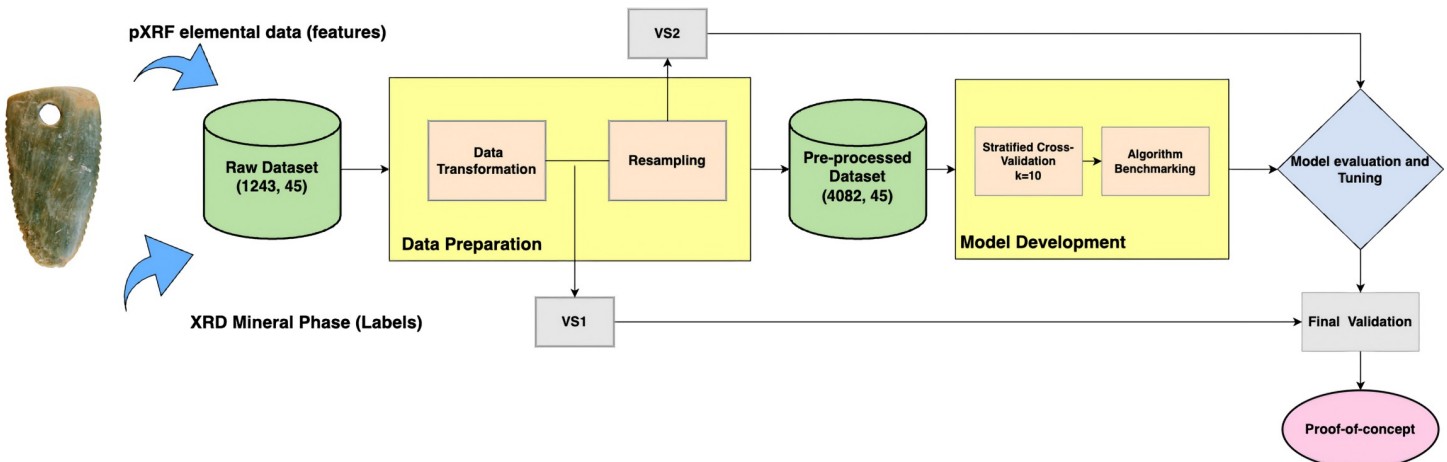

**Fig 2. Framework development: Schematic pipeline displaying steps in the development process.** The pre-processed dataset contains augmented data after resampling. Validation sets 1 and 2 (VS1 and VS2) were reserved at different stages for final validation before model deployment.

Jupyter notebooks demonstrating the development of the framework, and the proof-of-concept, as well as an option to use the trained model on user data and a static HTML with the diffractograms used as a ground truth. The algorithms can be obtained freely in the Scikit-Learn [43] and Pycaret [46] libraries.

## Results and discussion

For p-XRF measurements the mean analysis time per sample was 60 sec. (sd = 0.28) and all the chemical elements from Mg to Th were measured. Bead metrics were not used in this study.

Labels were obtained comparing the XRD diffractograms with the International Centre for Diffraction Data Powder Diffraction File database (ICDD-PDF2). In $\sim$16% ($n$ = 205) of the data, samples had more than one label assigned in which case we followed the criteria explained above.

All models were evaluated using the test set. The metrics used for evaluation are derived from the computation of the confusion matrix for binary classification problems that summarizes relevant information about the performance of the algorithms. From the identification of True Positives (TP), False Negatives (FN), False Positives (FP) and True Negatives (TN), measures of Accuracy, Recall, Precision and F-score can be calculated (Table 3). In a multi-class classification problem, these concepts must be adapted from the standpoint of each class [47]. Macro and weighted averages were taken into account to evaluate the models [48].

### Model benchmarking

Table 4 shows the comparison of the different models. The results are presented based on the ($k$ = 10) iteration average of the metrics defined above.

In general terms, decision tree-based models outperformed other classification logics. The top five best performers include this type of algorithm. A plausible explanation for this may be that they fit well with the structure of the data since in our case study, major mineral groups and species are discriminated by the presence or absence of some chemical elements or their relative percentage within the elemental composition. Thus, the data are structured hierarchically in decision nodes. Since the classification logic of the decision trees measures the information gain obtained at each node, which determines the decision-making process of the algorithm by calculating the relative importance of each chemical element within the classification process, the isomorphism between the structure of the data and the decision-making process of decision trees explains to a large extent the high performance of this type of models.

**Table 3. Evaluation metrics used in this study.**

| Measure | Evaluation focus |
|---|---|
| General Accuracy | Measures the probability that the predictions made by the model are correct. |
| Precision | Reports on the model's reliability when it classifies an item positively. |
| Recall | Reports the ability of the model to find the instances belonging to a class within the dataset. |
| F-score | Is the harmonic mean between recall and precision. Indicates the performance of the model in all classes. |

Macro methods tend to calculate the overall averages of the metrics, which implies that larger classes have the same weight as smaller classes in the calculation. Weighted methods consider the differential weight of each class. When dealing with imbalanced datasets, it is important to take these metrics into account as they provide us with differential information about the behaviour of the model.

**Table 4. Model benchmark.**

| Model | Tag | Accuracy | Recall | Prec. | F1 | TT (Sec) |
|---|---|---|---|---|---|---|
| Light Gradient Boosting Machine | lightgbm | 0.9850 | 0.9850 | 0.9857 | 0.9848 | 3.623 |
| Extra Trees Classifier | Et | 0.9822 | 0.9822 | 0.9831 | 0.9821 | 1.041 |
| Extreme Gradient Boosting | xgboost | 0.9807 | 0.9807 | 0.9814 | 0.9806 | 5.593 |
| Random Forest Classifier | Rf | 0.9780 | 0.9780 | 0.9789 | 0.9777 | 1.527 |
| Decision Tree Classifier | Dt | 0.9384 | 0.9384 | 0.9409 | 0.9382 | 1.044 |
| K Neighbors Classifier | knn | 0.9087 | 0.9087 | 0.9147 | 0.9069 | 0.903 |
| Logistic Regression | Lr | 0.8505 | 0.8505 | 0.8547 | 0.8457 | 2.139 |
| Naive Bayes | Nb | 0.7599 | 0.7599 | 0.7675 | 0.7200 | 0.919 |
| Linear Discriminant Analysis | lda | 0.7357 | 0.7357 | 0.7594 | 0.7289 | 0.855 |
| Ridge Classifier | ridge | 0.6750 | 0.6750 | 0.7079 | 0.6441 | 0.974 |
| SVM - Linear Kernel | svm | 0.5905 | 0.5905 | 0.6161 | 0.5457 | 1.061 |
| Quadratic Discriminant Analysis | qda | 0.5072 | 0.5072 | 0.6889 | 0.5036 | 0.896 |
| Ada Boost Classifier | ada | 0.1859 | 0.1859 | 0.1313 | 0.1218 | 1.167 |

Other models such as Knn also showed remarkable performance and are suitable for optimization as it is possible to control the number of neighbours in the model space by which the algorithm classifies each case. However, the chemical complexity of raw materials reflected in compositional data tends to form overlapping clusters that make it challenging to establish clear boundaries between classes when distance (e.g euclidian) is used as classification criterion, which negatively impacts the performance of models with this type of logic.

Models with other classification logics do not seem to fit the data structure and the nature of our case study as well. Thus, one of the most relevant results obtained corresponds to the confirmation of the great advantage of decision trees when classifying objects according to their chemical composition, since in addition to their remarkable performance, they allow a simple and transparent interpretation of their classificatory logic, in addition to their relatively low computational cost.

A boosting model (lightgbm) [42] was selected for further development because it has proven to be consistent across multiple comparison routines. However, the benchmarking step has been kept in the framework and different models could be selected. Table 5 shows the metrics of the model before and after hyperparameter optimization; the performance is approximately the same and the standard deviation (sd) shows a slightly higher uniformity over the validation iterations, reflecting a greater convergence of its generalization capacity. It is worth noting that optimization accounts for the most computation-intensive process within the model development and improvements in the performance were just marginal.

## Model evaluation on validation sets

Prior to deployment for real-world use, the resulting model was tested on the validation sets reserved from the beginning of the process (VS1 and VS2). It should be noted that these subsets are part of the training dataset and are therefore labelled data whose evaluation is synthesised in the metrics considered in this study (Table 6).

The model predictions are presented in two columns with categorical values representing the main mineral groups and mineral species labels in addition to a third column with the top three prediction possibilities according to their probabilistic score (S1 Table). In this way, the outputs of the model are transparent and the interpretation of the results is easy for non-specialists.

**Table 5. Model performance before and after optimisation.**

| Model | Parameters | Accuracy | Recall | Prec. | F1 |
|---|---|---|---|---|---|
| Selected Model (lgbm) | boosting_type = 'gbdt', class_weight = 'balanced' colsample_bytree = 1.0, importance_type = 'split',learning_rate = 0.1, max_depth = -1, min_child_samples = 20, min_child_weight = 0.001, min_split_gain = 0.0, n_estimators = 100, n_jobs = -1, num_leaves = 31',subsample = 1.0, subsample_for_bin = 200000, subsample_freq = 0 | 0.9838 | 0.9838 | 0.9844 | 0.9836 |
| sd | | 0.0087 | 0.0087 | 0.0083 | 0.0087 |
| Optimized Model (lgbm) | boosting_type = 'gbdt', class_weight = 'balanced', colsample_bytree = 1.0, feature_fraction = 0.8, importance_type = 'split', learning_rate = 0.3, max_depth = -1,min_child_samples = 100, min_child_weight = 0.001, min_split_gain = 0, n_estimators = 230, n_jobs = -1, num_leaves = 60, reg_alpha = 0.0005, reg_lambda = 1, subsample = 1.0, subsample_for_bin = 200000, subsample_freq = 0 | 0.9847 | 0.9847 | 0.9854 | 0.9846 |
| sd | | 0.0048 | 0.0048 | 0.0046 | 0.0048 |

Within the pipeline, the parameter (n_iter = 1000) that determines the exhaustiveness of the tuning algorithm in the search space produced some changes in the hyperparameter configuration that do not seem to have affected the behaviour of the model, so, to make the process less intensive, this parameter could be set to 10 or 50 iterations without the risk of sacrificing the model's capacity.

Close inspection of the results in VS1 showed that, in general, all cases were correctly classified. Low precision and f-1 scores in the berlinite, metavariscite, and planerite classes are due to FP of variscite samples. The unique case of talc was correctly classified, but a variscite case was labeled as talc, which affected the precision and f-1 score metrics within the class.

Of the 80 variscites in the subset 73 were correctly classified (91% Recall score), three were misclassified as metavariscites, two as berlinites, one as planerite and one as talc. This misbehavior is due to a limitation of the data gathering technique, as p-XRF is not able to measure light elements below Mg or molecular data, so it is not possible to differentiate between variscite and its polymorphs, for example, metavariscite (same chemical formula, different structure) or between hydrated and non-hydrated phosphates such as berlinite and variscite (difference on water content).

If we assume the three FN of variscites misclassified as metavariscites and two as berlinites as positive results since both species belong to the same major group of phosphates and metavariscites and variscites are polymorphs with the same elemental composition, then the overall accuracy in VS1 increases slightly. The oddest behavior occurs in the case of variscite classified as talc. The probabilistic prediction shows that the model assigned a 35% probability of being talc and a 15% probability of being variscite (S1 Table). This low confidence in the decision is quite informative and functions as a signal about the need to inspect problematic cases more thoroughly.

VS2 contained augmented data in minor classes. The metrics show an impressive model performance of 98% (Table 6). A close inspection of results reveals just a few cases of misclassification: one berlinite was classified as calcite, and one calcite was classified as berlinite. Two talcs were misclassified, one as clinochlore and one as variscite, and three variscites were labeled as crandallite, planerite, and calcite (S1 Fig). The slightly low recall metric in the variscite class (89%) is due to these three cases, two of which belong to the main group of phosphates (crandallite and planerite) and are often found together in geoarchaeological samples. The impressive metrics score highlights a distinct fact of the model's good performance: the

**Table 6. Evaluation metrics for VS1 and VS2.**

| Class | VS1 Prec. | VS1 Rec. | VS1 f1-s. | VS2 Prec. | VS2 Rec. | VS2 f1-s. |
|---|---|---|---|---|---|---|
| Aheylite | 1.0 | 1.0 | 1.0 | 1.0 | 1.0 | 1.0 |
| Annabergite | 1.0 | 1.0 | 1.0 | 1.0 | 1.0 | 1.0 |
| Aragonite | – | – | – | 1.0 | 1.0 | 1.0 |
| Berlinite | 0.5 | 1.0 | 0.66 | 0.94 | 0.94 | 0.94 |
| Calcite | 1.0 | 1.0 | 1.0 | 0.94 | 0.97 | 0.95 |
| Chlorite-serpentine | 1.0 | 1.0 | 1.0 | 0.93 | 1.0 | 0.96 |
| Clinochlore | 1.0 | 1.0 | 1.0 | 0.96 | 0.92 | 0.94 |
| Crandallite | 1.0 | 1.0 | 1.0 | 0.96 | 1.0 | 0.97 |
| Fluorite | 1.0 | 1.0 | 1.0 | 1.0 | 1.0 | 1.0 |
| Illite | – | – | – | 1.0 | 1.0 | 1.0 |
| Metavariscite | 0.25 | 1.0 | 0.4 | 1.0 | 1.0 | 1.0 |
| Muscovite | 1.0 | 1.0 | 1.0 | 1.0 | 1.0 | 1.0 |
| Planerite | 0.75 | 1.0 | 0.85 | 0.95 | 1.0 | 0.97 |
| Quartz | 1.0 | 1.0 | 1.0 | 1.0 | 1.0 | 1.0 |
| Strengite | 1.0 | 1.0 | 1.0 | 1.0 | 1.0 | 1.0 |
| Talc | 0.5 | 1.0 | 0.66 | 1.0 | 0.92 | 0.96 |
| Turquoise | 1.0 | 1.0 | 1.0 | 1.0 | 1.0 | 1.0 |
| Variscite | 1.0 | 0.91 | 0.95 | 0.96 | 0.89 | 0.92 |
| accuracy | 0.94 | 0.94 | 0.94 | 0.98 | 0.98 | 0.98 |
| macro avg | 0.87 | 0.99 | 0.90 | 0.98 | 0.98 | 0.98 |
| weighted avg | 0.97 | 0.94 | 0.95 | 0.98 | 0.98 | 0.98 |

Since VS1 was created before the resampling, the number of available instances of some classes was limited. Therefore, the Illite and Aragonite classes are not represented in this subset.

risk of using data augmentation. The strategy applied to deal with class imbalance resulted in the creation of homogeneous data points and therefore easy to classify correctly.

## Proof-of-concept results

The samples selected for the proof-of-concept correspond, as mentioned above, to two sites in the Portuguese Estremadura: Cova das Lapas and Gruta da Marmota. These samples correspond to real cases that have not been taken into account for model training and therefore accurately reflect the generalisation capacity of the model when faced unseen data (S1 File).

At the broadest level of classification, all samples were correctly classified as silicates (S1 Table). However, when attempting a more granular classification, the general accuracy was 76%, weighted precision, recall and f-1 score were 84%, 76% and $\sim$78% respectively (S2 Table).

All Muscovites were classified correctly as well as the two fluorites. Ten of the twelve Talcs were correctly classified, but one was misclassified as Clinochlore (CLP_141) and one as a Variscite (CLP_148). In both cases, the confidence level was relatively low ($\sim$ 52% for CLP_141 and $\sim$ 82% for CLP_148). Furthermore, CLP_148 was a unique case of discrepancy between the predictions at the two classification levels, correctly classified as a silicate and then as a Variscite, a type of phosphate. These behaviours when there is a discrepancy between predicted mineral groups and species and when the level of confidence is low (as discussed above) function as a composite result that enables a more informed decision making scenario by re-inspecting the input data, saving time and facilitating the workflow of systematic sampling in datasets with unknown or partially known mineralogy.

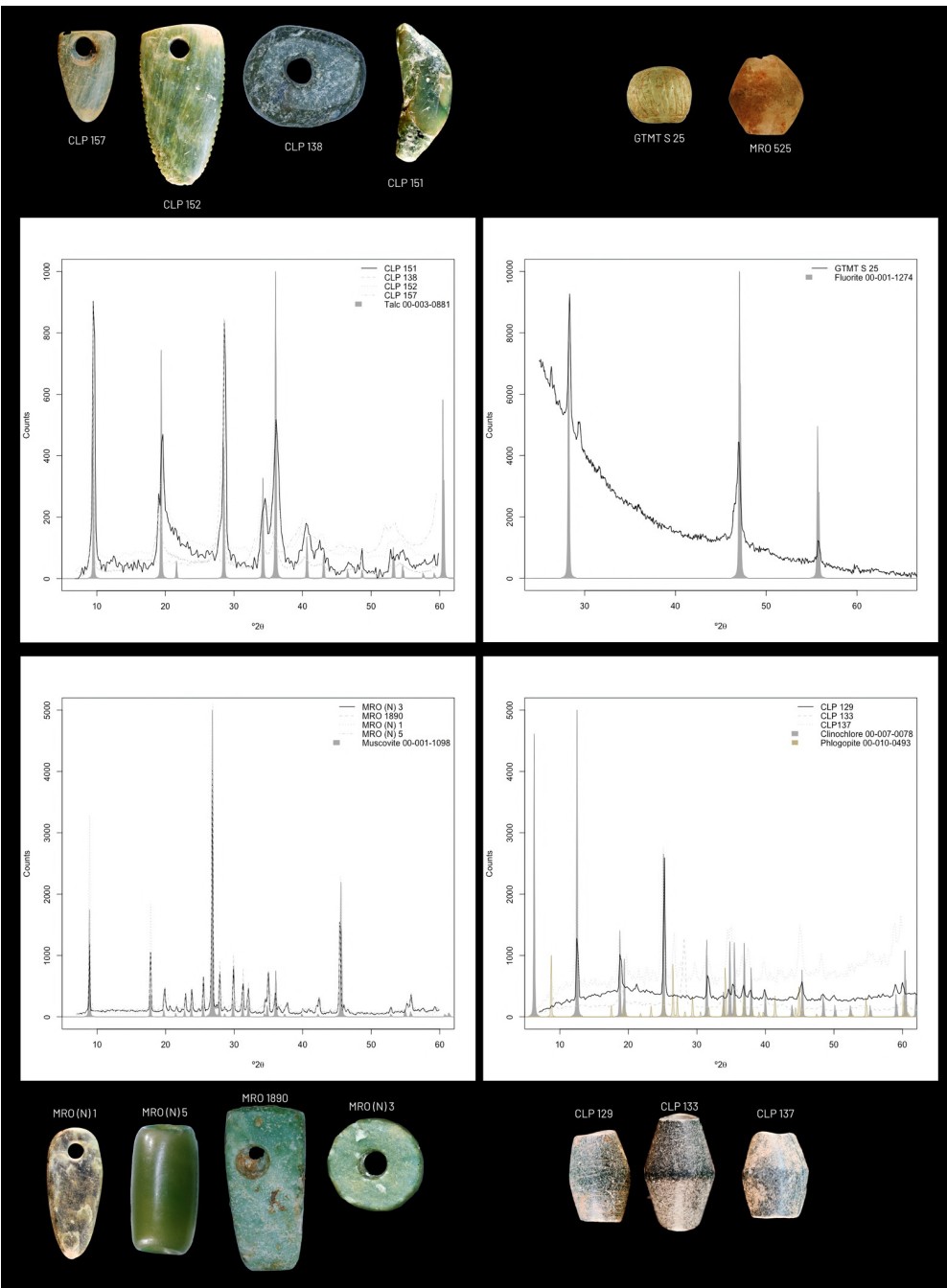

**Fig 3. Proof-of-concept samples photographic representation of the most representative materials used as proof of concept with their respective diffractograms used as ground truth for the evaluation of the model.** Note the process of interpretation of the diffractograms by comparing the characteristic peaks with those of the reference database. The rest of the diffractograms are provided in the S1 File. Talcs and clinochlores are from Cova das Lapas; Muscovites and fluorites are from Gruta da Marmota.

Four of the six clinochlores were misclassified, three as muscovite and one as chlorite-serpentine. Clinochlore is a member of the chlorite group that has been individualized in the class system due to the availability of sufficient occurrences in the training set ($n = 52$). For other

mineral species of the group, such as Cookeite, Chamosite, Lizardite, Antigorite or Chrysotile, we have decided to create a specific class (chlorite-serpentine) in order to group those cases that individually have few occurrences but as a whole have an important presence in the mineral composition of Iberian beads. Thus, although they are false negatives of the clinochlore class, the prediction was quite accurate given the belonging of this mineral phase to the chlorite group.

In the case of the three clinochlore misclassified as muscovites (CLP_133, CLP_136, CLP_137) (Fig 3), these samples contain a second label of a type of mica called Phlogopite, identified in the compositional data with about ∼10% K. Since the model has only one member of the mica group (Muscovite) in its class system, the amount of K has led to a decision for the only mica group member present in the model.

A striking aspect of this proof-of-concept is the considerable time advantage of using the method presented in this study. Although the samples used are part of the materials recovered from the excavation campaigns carried out by some team members, thus facilitating their loan and transfer, routine XRD analyses could take up to several hours including decision making and for 25 samples could take up to several days to reach a comprehensive result table. In comparison, compositional analysis by p-XRF takes an average of 60s per sample, after which can be used directly as input to the model. Therefore, in about ∼30 minutes we could obtain the same results. Furthermore, the outputs of the model not only provide an instantaneous classification of the samples, but also allow a transparent evaluation of the results that enables a quick identification of those samples that the model has had problems to classify. Thus the proposed method allows considerable time savings in two ways: by generating an immediate label from the data taken by p-XRF and by allowing the rapid identification of cases that need more careful analysis.

## Final remarks

The model's behaviour in the validation subsets has shown a tendency to overestimate its predictive capacity when facing synthetic data. This is evident when we compare the performance in both (∼98% accuracy on VS2 and ∼94% on VS1).This is most likely due to the low variance given the few cases available in the minority classes at the time of resampling; thus, synthetic data tend to be rather homogeneous and therefore easy to classify. To address this, a further step in the development of this research is to collect more data from the minority classes and retrain the model without using data augmentation.

Despite the high scores achieved during the model evaluation, the proof-of-concept draws a more realistic picture. The model performs remarkably well at the general level but achieves approximately ∼78% accuracy if we add up all the cases where we consider some approximation to the characterisation of the samples at the mineral species level (e.g. mineral polymorphs, or samples classified within closely related classes such as chlorite-serpentine and clinochlores).

General metrics such as accuracy, precision, f-1 score and recall are only useful for evaluation in development environments. In a real-world scenario, we are blind about the true generalization capacity of the model and to assess its real applicability it is necessary to take into account the level of uncertainty reduction in the task it is intended to solve; in our case, to assist in the mineral identification workflow. Therefore, in addition to the possibility of obtaining a label predicted by the model, it seems useful to interpret the predictions using the coincidence or non-coincidence at the two levels (general groups and mineral species) and the probabilistic confidence as a compound signal for a detailed inspection of the problematic cases. Ultimately, the model's predictions greatly facilitate the decision-making process when classifying a sample, allowing for a quick and easy-to-interpret inspection of the results.

We propose a workflow for the interpretation of the results that is being used by our team for the characterisation of new cases that will help us to map the distribution patterns of mineral adornments in the Iberian Peninsula; i) check that the predictions at both levels match ii) check the confidence of the probability of the prediction and set a confidence threshold (we propose $\sim 85\%$) to maintain the label generated by the model. If both levels do not match or the probability score is too low, label the sample as problematic and re-inspect its compositional data taking into account the information obtained through the model.

## Framework limitations and potentials

The main limitation of the model is its restricted mineral species class system ($n$ = 18 classes).

The chemical complexity of the raw materials used for bead production in Iberian prehistory is difficult to reflect from a modelling standpoint, as we would need enough cases of each poly-mineral used as a raw material to train a model capable of achieving decent performance in every possible class. However, like rock-forming minerals, we hypothesise that the mineralogical diversity of the raw materials used for bead making have a non-normal, long-tail distribution. This means that a small number of mineral species are largely responsible for the variability of the archaeological reality, so although we cannot classify all possible combinations of minerals, we can point to a reliable classification of the main mineral phases that make up the beads. The evaluation of two different assemblages of beads from two different sites shows that all the main mineral species that compose the beads fall within the class system of the model. Therefore, we consider that the data used to train the model, collected in a large part of the Iberian Peninsula, reliably reflects the mineralogical diversity of the archaeological reality of Iberian personal adornments.

The abundance and diversity of raw materials used for bead making depend, firstly, on their geological availability and, secondly, on access to these sources through complex social networks that need to be explored and modelled. Furthermore, among the great variety of raw materials available, the selection of certain types of rocks was driven by cultural criteria aimed at satisfying specific tastes and social purposes, as well as by those characteristics of the material that make it susceptible to be worked. All these natural and social filters contribute to reducing the complexity that was intended to be modelled in this study.

The cultural taste for specific raw materials with particular characteristics such as lustre, texture or green colour is not a particular phenomenon of the western Mediterranean. Other areas such as the south-west of the United States, Mexico, Colombia and the Caribbean present similar cases involving raw materials such as jade, variscite and turquoise [13, 49, 50]. The approach proposed in this study, (the combination of analytical techniques such as p-XRF +XRD) to acquire quality data to build supervised models with reliable predictive capacity has a great potential to include new classes such as Jade or lapislazuli and contribute to addressing questions related to the consumption of this type of objects in a wide cultural and geographical range.

A further step in the applicability of the proposed model is its forthcoming integration into our PEPAdb (Prehistoric Europe's Personal Adornment Database) project [51] which analyses and publishes data related to personal adornments following the FAIR policies. In this way, relevant researchers in the field and other stakeholders will have access to and be able to contribute to knowledge creation in an open science approach, using the model to classify their own samples and expanding the database to address emerging issues related to personal ornamentation across a wide geographical scope. Other collaborative projects such as [52] can also benefit from research outcomes that include our model as part of their workflow, generating interesting

synergies in the growing trend of research interested in studying archaeological macro-patterns in large geographic regions involving the management of emerging amounts of data.

The casuistry of sites with personal adornment in the recent prehistory of the Iberian Peninsula is very diverse, ranging from large funerary centres, megalithic monuments and fortified settlements whose frequency of occurrence fluctuates from a few to hundreds or thousands [11, 13, 29]. Tools such as the one presented have the potential to streamline the collection of data from these sites, whether in situ as part of new excavations or sampling campaigns in museums and academic centres, which will undoubtedly contribute to advancing the questions that remain in this regard.

## Machine learning approaches to predict mineralogy on personal adornments

Geochemical methods have been successfully integrated into archaeological research in recent decades and have proven to be of great value when complemented by ML techniques. However, there are still certain challenges that limit wider integration, including the costs associated with the implementation of specific analytical techniques to acquire quality data and the expertise required to interpret the results.

Within the scope of this study, we consider that the framework developed contributes to filling a gap in the study of personal adornment by providing a tool that contributes to the rapid collection of empirical data while significantly reducing the time and costs associated with these challenges. P-XRF is a widely used and inexpensive technique, and the data obtained are easy to interpret and allow the construction of data matrices well suited to the requirements of some of the most versatile and least computationally intensive, yet effective, ML algorithms. Thus contributing to their reproducibility and reusability. Furthermore, geochemical data are constantly emerging in different domains and open-access initiatives are becoming increasingly widespread in this field, facilitating the acquisition of data that can be easily adapted for use in ML approaches applied to the archaeological domain.

Our framework exploits these advantages through an approach using decision tree-based algorithms that allow a transparent inspection of their decision-making process, thus facilitating the interpretation of the results and paving the way for mapping the regional distribution of beads of different mineralogies. (Fig 4).

## Conclusions

The archaeology of personal adornments in much of Western Europe, particularly on the Iberian Peninsula, lacks sufficient empirical data on raw material composition to reliably trace provenance and model the complex social networks associated with their consumption. In this paper we present an alternative method to overcome some of the challenges in acquiring quality data.

The primary goal of this work was to investigate the applicability of an ML approach for assisting mineral identification on prehistoric personal adornments. To accomplish this, we created the largest geochemical Iberian dataset to date, coupling compositional and molecular data from two of the most used spectroscopy techniques; pXRF and XRD.

We have trained and evaluated the performance of different supervised algorithms and developed an end-to-end framework that assists the mineral classification of prehistoric beads. Finally, we conducted a proof-of-concept with 25 archaeological samples of two unpublished Portuguese sites: Cova das Lapas and Gruta da Marmota.

The results obtained demonstrate that supervised approaches are suitable to assist the mineral classification workflow, when specific analyses are not available, allowing for fast and

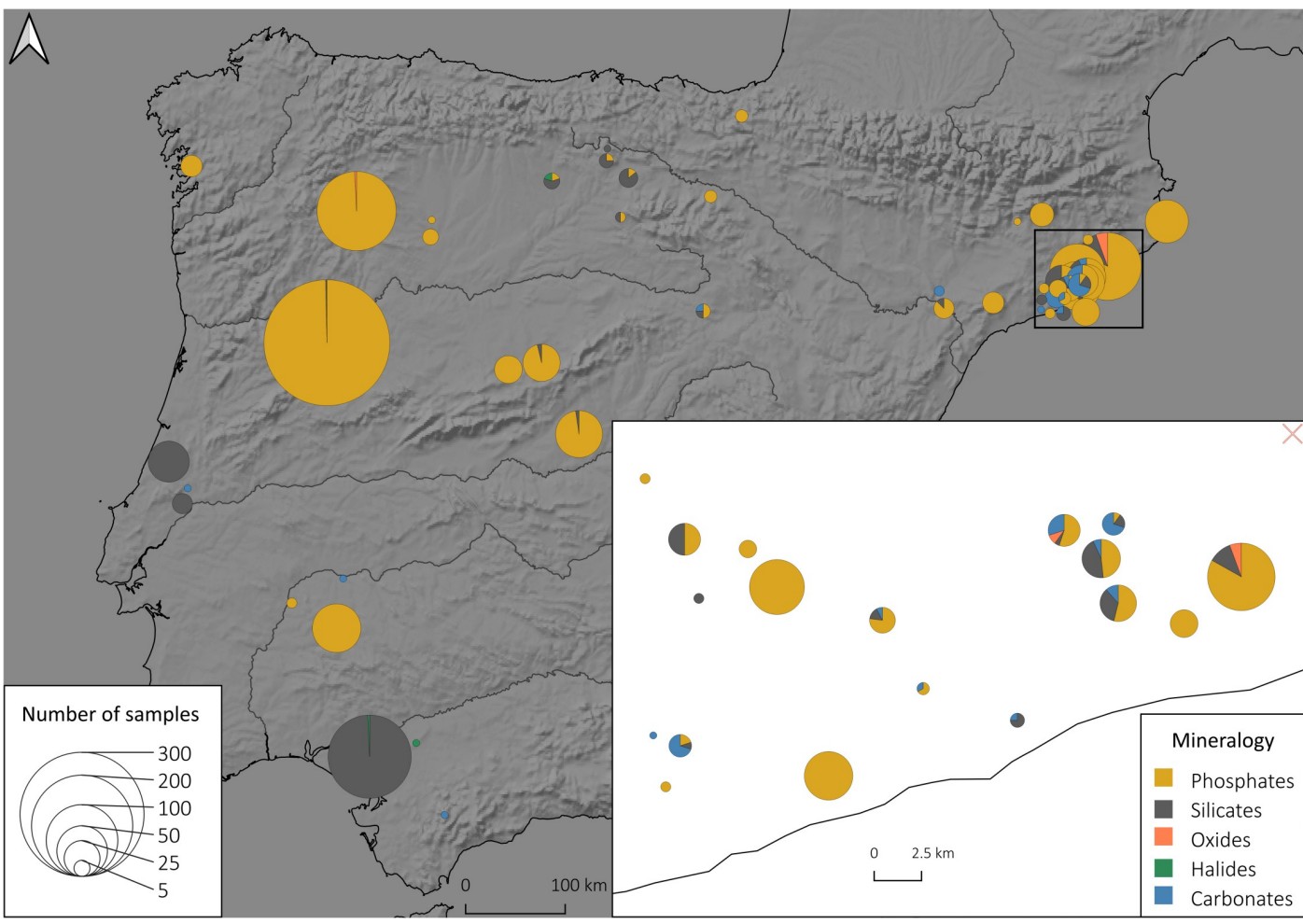

**Fig 4. Spatial distribution of general mineral groups used in this study: Mapping the distribution of beads from different mineralogies stimulates new research questions about the complex social networks related to these goods.**

reliable empirical data collection. We propose a workflow for the interpretation of model results that reduces uncertainty and facilitates systematic sampling.

Decision-trees-based classifiers seem to be the best suited for this type of classification task probably due to the hierarchical nature of data structure given the discriminatory importance of some chemical elements. Furthermore, its classification logic is quite transparent and allows for a clear assessment of the model and explanation of the results.

Finally, special attention has been paid in this work to present all the steps followed during the development of the model in an open and reproducible way, without the need to install any piece of software. Anyone can access the data and the code and reproduce the experiments, modify them, develop a new model or run the pre-trained models from a standard web browser.

## Supporting information

**S1 Dataset. Training dataset.** Data used to develop the models presented in this study. It contains eight tabs with training and proof-of-concept data, in addition to data and statistics of the reference material, detection limits, imputation values and number of samples of each site considered.
(XLSX)

**S1 Appendix. Dataset description.** PDF document with specific information about sample location, feature description and tab captions.
(PDF)

**S1 Fig. Proof-of-concept confusion matrix.**
(TIFF)

**S1 Table. Model predictions.** This table reports the model results for the different validation subsets and for the proof of concept. The VS1_predictions and VS2_predictions tabs inform about the predictions for the two validation subsets reserved at different stages of the model development process. VS1 corresponds to a subset reserved prior to the resampling process. VS2 contains resampled (i.e. synthetic) data. The tab proof_of_concept_predictions reports the results obtained for the use case presented in this study. The results within thi tab correspond to data never seen by the model and are therefore assumed to be results of the actual generalisation capacity of the model.
(XLSX)

**S2 Table Proof-of-concept evaluation metrics Accuracy, recall, precision and F-1 score metrics.**
(XLSX)

**S1 File. Ground truth.** XRD Diffractograms used as a baseline to evaluate the results of the proof-of-concept.
(PDF)

## Acknowledgments

We are especially grateful to all the researchers who have contributed in any way to the immense data collection effort over more than a decade, Primitiva Bueno-Ramirez; Rosa M Barroso Bermejo; Joao Luis Cardoso, José Antonio Linares-Catela, Joao Carlos Caninas and, Victor Hurtado Perez, Ferran Borrell, Germán Delibes, Special thanks to the Museu Municipal de Torres Novas and CRIVARQUE (Torres Novas, Portugal) Museu dos Serviços Geológicos de Portugal (Lisbon, Portugal), Museo de Huelva (Huelva, Spain), CIPAG, Museu Nacional de Arqueología de Catalunya (Barcelona, Spain) (Sevilla, Spain); Seminari d'Estudis I Recerques Prehistòriques (University of Barcelona, Spain); Filipa Rodrigues (CRIVARQUE), Museu de Bellas Artes de Castellón (Castellón de la Plana, Spain), Departamento de Prehistoria y Arqueología (Universidad de Sevilla, Spain), for enabling us to analyze and photograph the pieces in their collections and providing staff to assist in the fieldwork.

We would like to aknowledge to the academic editor Barry molloy and the two anonympus reviewers. Their detailed and incisive reading as well as their valuable insights has allowed to improve the manuscript.

## Author Contributions

**Conceptualization:** Daniel Sanchez-Gomez, Carlos P. Odriozola Lloret.

**Data curation:** Daniel Sanchez-Gomez, Ana Catarina Sousa, José Ángel Garrido-Cordero, Galo Romero-García, José María Martínez-Blanes, Manel Edo I. Benaiges, Victor S. Gonçalves.

**Formal analysis:** Daniel Sanchez-Gomez, Carlos P. Odriozola Lloret, José María Martínez-Blanes.

**Funding acquisition:** Carlos P. Odriozola Lloret, Ana Catarina Sousa.

**Investigation:** Daniel Sanchez-Gomez, Carlos P. Odriozola Lloret, Ana Catarina Sousa, José Ángel Garrido-Cordero, Galo Romero-García, José María Martínez-Blanes, Manel Edo I. Benaiges, Rodrigo Villalobos-García, Victor S. Gonçalves.

**Methodology:** Daniel Sanchez-Gomez, Carlos P. Odriozola Lloret, Ana Catarina Sousa, José Ángel Garrido-Cordero, Galo Romero-García, José María Martínez-Blanes.

**Project administration:** Daniel Sanchez-Gomez.

**Resources:** Daniel Sanchez-Gomez, Carlos P. Odriozola Lloret, Manel Edo I. Benaiges, Rodrigo Villalobos-García, Victor S. Gonçalves.

**Software:** Daniel Sanchez-Gomez, Galo Romero-García.

**Supervision:** Carlos P. Odriozola Lloret, Ana Catarina Sousa.

**Validation:** Carlos P. Odriozola Lloret, Ana Catarina Sousa, Galo Romero-García.

**Visualization:** Galo Romero-García.

**Writing – original draft:** Daniel Sanchez-Gomez.

**Writing – review & editing:** Daniel Sanchez-Gomez, Carlos P. Odriozola Lloret, Ana Catarina Sousa, José Ángel Garrido-Cordero, Manel Edo I. Benaiges, Rodrigo Villalobos-García.

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
