## [Decision Letter · Decision Letter 0]

18 Jan 2024

PONE-D-23-39546A supervised multiclass framework for mineral classification of Iberian beadsPLOS ONE

Dear Dr. Sanchez-Gomez,

Thank you for submitting your manuscript to PLOS ONE. After careful consideration, we feel that it has merit but does not fully meet PLOS ONE’s publication criteria as it currently stands. Therefore, we invite you to submit a revised version of the manuscript that addresses the points raised during the review process.

As you will see, the reviewers are generally positive about the analytical integrity of the work and your underlying technical methodology, but highlight several significant issues with the manuscript in its current form. One of their most substantial concerns is the applicability of this research both in terms of extracting social meaning with regard to your own dataset and with regard to the further applications of the method and model developed herein. In particular, Reviewer 1 draws attention to your own uncertainty about what new insights the method has brought in terms of archaeological interpretation of the dataset. This calls for a substantial reconsideration of how you present your analyses, results and their wider relevance beyond this specific exercise. Sentences that Reviewer 1 highlights, but more generally the tone of the narrative, that imply your results are “wrong” may benefit from reconsidering how you frame your evaluation if you believe your results have value and are more than simply “wrong”. The value in addressing archaeological questions with the aid of this model should be more coherently justified. Overall, this study has potential and a substantial reframing of its relevance and potential impact alongside addressing specific reviewer concerns will be required before reconsideration for publication.

Please pay particular attention to required changes. These include demonstrating how your results are meaningful and as well as rephrasing, explaining more clearly what you mean when you say that aspects of the results were wrong and the means through which this was overcome within your model. 

You must also clarify the relevance and potential impact of the model as a stand alone approach. If the opinion of the analyst remains paramount to the final outcome, in which ways has the model contributed to understanding? This requires a reconsideration of the context and potential implementation of the model developed and overall, how this is presented. 

The real-world utility of the model needs to be considered with regard to studies of various scales, from individual field projects to museum-based projects using larger datasets. 

The datasets / spreadsheets need more complete captioning, explanation, labelling and to be more coherently integrated into the discussion of analyses (e.g. the number of objects ultimately studied).

We look forward to receiving your revised manuscript.

Kind regards,

Barry Molloy

Academic Editor

PLOS ONE

2. In your manuscript, please provide additional information regarding the specimens used in your study. Ensure that you have reported human remain specimen numbers and complete repository information, including museum name and geographic location. 

For more information on PLOS ONE's requirements for paleontology and archeology research, see https://journals.plos.org/plosone/s/submission-guidelines#loc-paleontology-and-archaeology-research.

 [This research has received resources from the Fundação para a Ciência e Tecnologia de Portugal (www.fct.pt) in the framework of the doctoral grant (UI/BD/154365/2023) awarded to (DSG) and from the Spanish Ministry of Science and Technology through the research project (PID2021-124421NB-I00) (https://investigacion.us.es/sisius/sis_proyecto.php?idproy=33567) whose PI is (CO) and the research team is composed by (JMMB), (GRG), and (JAGC). The other authors report no funding for participation in this project. ].  

5. We note that Figure 1 and 4 in your submission contain [map/satellite] images which may be copyrighted. All PLOS content is published under the Creative Commons Attribution License (CC BY 4.0), which means that the manuscript, images, and Supporting Information files will be freely available online, and any third party is permitted to access, download, copy, distribute, and use these materials in any way, even commercially, with proper attribution. For these reasons, we cannot publish previously copyrighted maps or satellite images created using proprietary data, such as Google software (Google Maps, Street View, and Earth). For more information, see our copyright guidelines: http://journals.plos.org/plosone/s/licenses-and-copyright.

a. You may seek permission from the original copyright holder of Figure 1 and 4 to publish the content specifically under the CC BY 4.0 license.  

6. We note that Figure 2 and 3 in your submission contain copyrighted images. All PLOS content is published under the Creative Commons Attribution License (CC BY 4.0), which means that the manuscript, images, and Supporting Information files will be freely available online, and any third party is permitted to access, download, copy, distribute, and use these materials in any way, even commercially, with proper attribution. For more information, see our copyright guidelines: http://journals.plos.org/plosone/s/licenses-and-copyright.

a. You may seek permission from the original copyright holder of Figure 2 and 3 to publish the content specifically under the CC BY 4.0 license. 

Academic Editor Comments:

In your manuscript, please provide additional information regarding the specimens used in your study. Ensure that you have reported specimen numbers and complete repository information, including museum name and geographic location.

For more information on PLOS ONE's requirements for paleontology and archaeology research, see https://journals.plos.org/plosone/s/submission-guidelines#loc-paleontology-and-archaeology-research.

2) We note that Figures 1 and 4 in your submission contain [map/satellite] images which may be copyrighted. All PLOS content is published under the Creative Commons Attribution License (CC BY 4.0), which means that the manuscript, images, and Supporting Information files will be freely available online, and any third party is permitted to access, download, copy, distribute, and use these materials in any way, even commercially, with proper attribution. For these reasons, we cannot publish previously copyrighted maps or satellite images created using proprietary data, such as Google software (Google Maps, Street View, and Earth). For more information, see our copyright guidelines: http://journals.plos.org/plosone/s/licenses-and-copyright.

a. You may seek permission from the original copyright holder of Figures 1,3,4,5,8,9 and 12 to publish the content specifically under the CC BY 4.0 license.

3) Please provide details on copyright permissions for all images of objects reproduced in the figures and for all illustrations.

Reviewers' comments:

Reviewer's Responses to Questions

**Comments to the Author**

1. Is the manuscript technically sound, and do the data support the conclusions?

Reviewer #1: Yes

Reviewer #2: Yes

2. Has the statistical analysis been performed appropriately and rigorously? 

Reviewer #1: Yes

Reviewer #2: Yes

3. Have the authors made all data underlying the findings in their manuscript fully available?

Reviewer #1: Yes

Reviewer #2: Yes

4. Is the manuscript presented in an intelligible fashion and written in standard English?

Reviewer #1: Yes

Reviewer #2: Yes

5. Review Comments to the Author

Reviewer #1: The main aim of the study presented in this manuscript is to use machine-learning algorithms to achieve mineralogical classifications of prehistoric beads. The use of this computational technique is presented as suitable in cases where the application of analytical methods other than portable XRF is not feasible. In general, the aim and execution of the exercise is original and potentially innovative, as it presents a new way to look at the classification of beads in terms of mineralogical composition. However, there remain serious doubts as to the overall efficiency and meaningfulness of the results.

To begin with, in page 15 of the manuscript, the authors themselves admit that “the chemical complexity of raw materials reflected in compositional data tends to form overlapping clusters that make it challenging to establish clear boundaries between classes, which obscures the process of model selection with this type of logic”. Later, in page 19, they go on to admit that “even if the results are strictly wrong, they are very close to an accurate characterization of the samples”. This is a remarkable statement in a number of ways. The authors claim that their results are wrong. The same idea is conveyed further on, in page 27: “Thus, though strictly wrong, the prediction was quite accurate”.

These statements raise question as to how the results of the computational classification are assessed or verified. How do the authors know the results are wrong? No explanation is provided of the criteria by which the results are assessed. And then, if the results are wrong, what is the point of the exercise, anyway? Immediately after admitting that the results are wrong they claim that “they are very close to an accurate description of the samples” (page 19). If the authors have a pre-ordained understanding of what is the right and wrong classification, what is the point of a computerised model that provides a result deemed essentially wrong? And how do they know it is wrong?

The confusion stemming from these statements is further compounded in page 29, when it is claimed that “Ultimately, the model’s predictions may nicely complement the decision-making process of labelling a sample, which is, after all, up to the analyst”. A clearer definition of the intended meaning of “labelling” is also required here, but if we take it to mean “classification”, then the inevitable question is: if the task of classifying the beads according the mineralogy lies basically with the analyst, what is the point of the whole computational model?

A much clearer justification of the validity and worthiness of the whole exercise is required, because otherwise it might appear a futile.

Minor corrections

Line 114: “for the identification of bead-forming minerals” sounds a bit odd, as if the minerals had an “intention” to form beads.

Line 119: where it says “two sample sets of different sites” it should say “two sample sets from different sites”

Line 125: it should say “millennia”

Line 277: where it says “our problem” it should say “our case study”

Line 305: the meaning of the term “pipeline” in the context of machine-learning algorithms should be explained, as readers not familiar with these computational models will not be familiar with it.

Line 358: The whole paragraph is duplicated

Line 376. Table 7 is too long. It should be presented as supplementary data.

Line 459: “bead carving” should be “bead making”

Line 468: This paragraph should go much earlier on in the paper, in the introduction.

Reviewer #2: The study tests and explores the potentials of using a machine learning / multiclass model for the identification of minerals in beads from the Iberian Peninsula.

The text is well written. The authors present the protocols and methodology in detail, and importantly the limitations for the algorithm.

Archaeology is still in the rather early stages of adopting machine learning methods widely (compared to other fields), and in this context the study has value. However, certain aspects of the paper are not clear. I would advise reconsideration of a few aspects before publication.

Firstly, the paper mentions a database of more than 1k objects, but in reality, a much smaller sample is used for this proof-of-concept study case. I understand that the skewed sample (with ca. 800 variscite beads) needs to be adjusted for training the model. However, as of now, my impression is that the 1.2k bead database is not relevant to the resulting model, and I do not see why it should be mentioned in the paper. The authors should highlight which are (if any) the contributions of the 1.2k bead database in the success of the model. Otherwise, I would exclude this aspect altogether.

Secondly, even though the authors present a road map for using their model, my general critique is that I would like to see suggestions for a real-life scenario where the applicability of the model by relevant researchers in the field would be realistic and timewise sensible. For example, how many such ornaments are expected to be found on a typical site? Would this number justify the use of the model (which would require a specialised person)? In the hypothetical case of an assemblage of some 30 beads from a site, wouldn’t it be possible to scan them with the pXRF and gauge their mineralogy based on the compositions? Which is what is also suggested by the authors in the case of doubtful results. If this model is destined for large museum collections, could this proof-of-concept be tested against the whole 1.2k database (and include the results in the publication)?

Finally, it is not clear if this model is trained only for the Iberian Peninsula and / or only for this type of objects. Would the model be able to identify an unknown material (insteady of mislabelling it)? Please add specific information about the spectrum of applicability of this ML trained model.

Additional comments on supplementary data provided:

S1. Database:

1. Each tab needs a table caption explaining the variables and info included in the tab. Indicate if elements are presented as such or as compounds (considering the analysis of minerals) and justify your choice.

2. Numbers are stored as text making it difficult to judge the data. Please convert to numbers.

3. Values below the detection limits are not indicated. Only include the significant decimal places (pXRF will give many decimals that don't mean much).

4. Include the analysis of the standard materials with statistical information about their precision and accuracy, and respective detection limits.

5. In the proof-of-concept tab, the xrd3 column is empty – why is that?

S2. Ground truth – Add text that explains the graphs.

As a bottom line, I would encourage the authors to add some more emphasis on the real-life applicability of this model and pay some more attention to the treatment and presentation of the database of pXRF values.

6. PLOS authors have the option to publish the peer review history of their article (what does this mean?). If published, this will include your full peer review and any attached files.

Reviewer #1: No

Reviewer #2: No

---

## [Author Response · Author response to Decision Letter 0]

29 Feb 2024

We would like to acknowledge the valuable and detailed readings, recommendations and revisions made to the manuscript by both the academic editor and the reviewers. Their reviews and criticisms have been very accurate and have allowed us to improve the quality of the work. We hope that this new version will elicit a positive response from them.

We have considered all comments and addressed most of them within the revised version. In this letter, we provide responses (in blue) to each of the raised points.

Sincerely,

Daniel Sanchez-Gomez

Corresponding author (on behalf of all co-authors),

To the academic editor,

As you will see, the reviewers are generally positive about the analytical integrity of the work and your underlying technical methodology, but highlight several significant issues with the manuscript in its current form. One of their most substantial concerns is the applicability of this research both in terms of extracting social meaning with regard to your own dataset and with regard to the further applications of the method and model developed herein. In particular, Reviewer 1 draws attention to your own uncertainty about what new insights the method has brought in terms of archaeological interpretation of the dataset. This calls for a substantial reconsideration of how you present your analyses, results and their wider relevance beyond this specific exercise. Sentences that Reviewer 1 highlights, but more generally the tone of the narrative, that imply your results are “wrong” may benefit from reconsidering how you frame your evaluation if you believe your results have value and are more than simply “wrong”. The value in addressing archaeological questions with the aid of this model should be more coherently justified. Overall, this study has potential and a substantial reframing of its relevance and potential impact alongside addressing specific reviewer concerns will be required before reconsideration for publication.

Please pay particular attention to required changes. These include demonstrating how your results are meaningful and as well as rephrasing, explaining more clearly what you mean when you say that aspects of the results were wrong and the means through which this was overcome within your model.

You must also clarify the relevance and potential impact of the model as a stand alone approach. If the opinion of the analyst remains paramount to the final outcome, in which ways has the model contributed to understanding? This requires a reconsideration of the context and potential implementation of the model developed and overall, how this is presented. 

The real-world utility of the model needs to be considered with regard to studies of various scales, from individual field projects to museum-based projects using larger datasets. 

The datasets / spreadsheets need more complete captioning, explanation, labelling and to be more coherently integrated into the discussion of analyses (e.g. the number of objects ultimately studied).

R:

We have reworded, modified and expanded some paragraphs and renamed some sections to improve coherence and further explain the potential and relevance of the method presented. In particular we have:

Constructed a new paragraph in the introduction section (line 72) highlighting the challenges derived from the lack of empirical data in the study of prehistoric social dynamics related to the consumption of personal adornment in Western Europe.

In line 341 we have built a new paragraph highlighting the advantage of decision trees over other types of classificatory models for cases such as ours (classification from chemical composition)

In line 375 we have constructed a new paragraph explaining more clearly how the results in the validation sets were evaluated, emphasising that as the data used for the model development, they are labelled and their evaluation is easy from the standpoint of the metrics considered in the study.

In line 443 we have renamed the "proof-of-concept" section to "proof-of-concept results".

In line 503 we have renamed the “framework limitations” section to “ framework limitations and potentials, highlighting the potential applications of the model and its upcoming integration in our recently published PEPAdb (Prehistoric Europe’s Personal Adornment Database) project.(García et al., 2023)

In line 482 we have constructed a paragraph explaining more clearly the great time advantage of using our approach to classify personal ornaments.

Other minor changes throughout the document have provided greater consistency, highlighted the advantages of the model in projects of different scales, and clarified concerns about the term "wrong" used to describe some results.

We have also constructed an additional supplementary information file (S1.Appendix. dataset description) in response to suggestions for a more detailed description of the dataset. This document contains specific information on the location of the specimens used, as well as data and statistics on reference materials.

All requirements of the journal and the academic editor regarding copyright and permissions on specimens used have been addressed.

We have considered all the issues raised and have provided answers to each of them for both reviewers. We consider that the changes we have made have improved the quality of the text and addressed their concerns. We hope that the revised version fits their expectations better.

2. In your manuscript, please provide additional information regarding the specimens used in your study. Ensure that you have reported human remain specimen numbers and complete repository information, including museum name and geographic location. 

For more information on PLOS ONE's requirements for paleontology and archeology research, see https://journals.plos.org/plosone/s/submission-guidelines#loc-paleontology-and-archaeology-research.

R:

Within the manuscript in the “Data and code availability” (line 281) we state: No human remains were used in the present study. No permits were required for the analysis of geological samples. Permits for the analyses of archaeological samples were granted by the Chief Curators and Heads of the museums listed as supporting information in (S1. Appendix. Dataset description) as well as specific information about sample location.

R:

We have amended the information in the corresponding section.

 [This research has received resources from the Fundação para a Ciência e Tecnologia de Portugal (www.fct.pt) in the framework of the doctoral grant (UI/BD/154365/2023) awarded to (DSG) and from the Spanish Ministry of Science and Technology through the research project (PID2021-124421NB-I00) (https://investigacion.us.es/sisius/sis_proyecto.php?idproy=33567) whose PI is (CO) and the research team is composed by (JMMB), (GRG), and (JAGC). The other authors report no funding for participation in this project. ]. 

R:

We have amended this information in the cover letter.

5. We note that Figure 1 and 4 in your submission contain [map/satellite] images which may be copyrighted. All PLOS content is published under the Creative Commons Attribution License (CC BY 4.0), which means that the manuscript, images, and Supporting Information files will be freely available online, and any third party is permitted to access, download, copy, distribute, and use these materials in any way, even commercially, with proper attribution. For these reasons, we cannot publish previously copyrighted maps or satellite images created using proprietary data, such as Google software (Google Maps, Street View, and Earth). For more information, see our copyright guidelines: http://journals.plos.org/plosone/s/licenses-and-copyright.

R:

Figure 1 and 4 base maps are from Natural Earth (public domain):

https://www.naturalearthdata.com/about/terms-of-use/

“All versions of Natural Earth raster + vector map data found on this website are in the public domain. You may use the maps in any manner, including modifying the content and design, electronic dissemination, and offset printing. The primary authors, Tom Patterson and Nathaniel Vaughn Kelso, and all other contributors renounce all financial claim to the maps and invites you to use them for personal, educational, and commercial purposes.

No permission is needed to use Natural Earth. Crediting the authors is unnecessary.

However, if you wish to cite the map data, simply use one of the following.

Short text: Made with Natural Earth.

Long text: Made with Natural Earth. Free vector and raster map data @ naturalearthdata.com.”

6. We note that Figure 2 and 3 in your submission contain copyrighted images. All PLOS content is published under the Creative Commons Attribution License (CC BY 4.0), which means that the manuscript, images, and Supporting Information files will be freely available online, and any third party is permitted to access, download, copy, distribute, and use these materials in any way, even commercially, with proper attribution. For more information, see our copyright guidelines: http://journals.plos.org/plosone/s/licenses-and-copyright.

R: 

Figure 2 has been created by means of drawio.com https://www.drawio.com

“There are no restrictions on the usage of any diagrams you generate with draw.io, including our online editor at app.diagrams.net. The diagrams you create belong to you and we grant you a license to use any of our copyrighted icons in your diagrams and for making raster and vector versions of your diagrams and parts, using draw.io, for any purpose.

Can they be used for any purpose under commercial usage conditions, for example?

Yes

Does that extend to using the built-in icons that are copyrighted to JGraph?

Yes”

https://www.drawio.com/doc/faq/usage-terms

For figure 3 we are the creators, both the images and the x/y graphic, so they can be freely used under the Creative Commons Attribution License (CC BY 4.0).

Academic Editor Comments:

In your manuscript, please provide additional information regarding the specimens used in your study. Ensure that you have reported specimen numbers and complete repository information, including museum name and geographic location.

For more information on PLOS ONE's requirements for paleontology and archaeology research, see https://journals.plos.org/plosone/s/submission-guidelines#loc-paleontology-and-archaeology-research.

R:

No human remains of any kind were used in this study.

Specific Information regarding sample locations and their availability to qualified researchers is available in the supplementary material (S1.Appendix. Dataset description).

2) We note that Figures 1 and 4 in your submission contain [map/satellite] images which may be copyrighted. All PLOS content is published under the Creative Commons Attribution License (CC BY 4.0), which means that the manuscript, images, and Supporting Information files will be freely available online, and any third party is permitted to access, download, copy, distribute, and use these materials in any way, even commercially, with proper attribution. For these reasons, we cannot publish previously copyrighted maps or satellite images created using proprietary data, such as Google software (Google Maps, Street View, and Earth). For more information, see our copyright guidelines: http://journals.plos.org/plosone/s/licenses-and-copyright.

R:

Figure 1 and 4 base maps are from Natural Earth (public domain):

https://www.naturalearthdata.com/about/terms-of-use/

“All versions of Natural Earth raster + vector map data found on this website are in the public domain. You may use the maps in any manner, including modifying the content and design, electronic dissemination, and offset printing. The primary authors, Tom Patterson and Nathaniel Vaughn Kelso, and all other contributors renounce all financial claim to the maps and invites you to use them for personal, educational, and commercial purposes.

No permission is needed to use Natural Earth. Crediting the authors is unnecessary.

However, if you wish to cite the map data, simply use one of the following.

Short text: Made with Natural Earth.

Long text: Made with Natural Earth. Free vector and raster map data @ naturalearthdata.com.”

a. You may seek permission from the original copyright holder of Figures 1,3,4,5,8,9 and 12 to publish the content specifically under the CC BY 4.0 license.

Reviewers' comments:

5. Review Comments to the Author

Reviewer #1: The main aim of the study presented in this manuscript is to use machine-learning algorithms to achieve mineralogical classifications of prehistoric beads. The use of this computational technique is presented as suitable in cases where the application of analytical methods other than portable XRF is not feasible. In general, the aim and execution of the exercise is original and potentially innovative, as it presents a new way to look at the classification of beads in terms of mineralogical composition. However, there remain serious doubts as to the overall efficiency and meaningfulness of the results.

To begin with, in page 15 of the manuscript, the authors themselves admit that “the chemical complexity of raw materials reflected in compositional data tends to form overlapping clusters that make it challenging to establish clear boundaries between classes, which obscures the process of model selection with this type of logic”. Later, in page 19, they go on to admit that “even if the results are strictly wrong, they are very close to an accurate characterization of the samples”. This is a remarkable statement in a number of ways. The authors claim that their results are wrong. The same idea is conveyed further on, in page 27: “Thus, though strictly wrong, the prediction was quite accurate”.

R:

We would like to acknowledge the reviewer's careful reading of our manuscript. The wise questions and points highlighted have made us pay more attention to the way we present the details of our research and the proposed method. We have paid special attention to the terms used to present th

---

## [Decision Letter · Decision Letter 1]

27 Mar 2024

PONE-D-23-39546R1A supervised multiclass framework for mineral classification of Iberian beadsPLOS ONE

Dear Dr. Sanchez-Gomez,

Thank you for re-submitting your manuscript to PLOS ONE. After careful consideration, we feel that it has merit but still does not fully meet PLOS ONE’s publication criteria as it currently stands. Therefore, we invite you to submit a revised version of the manuscript that addresses the points raised during the review process.

Thank you for your revised manuscript and for your careful attention to the comments by the reviewers and the editor of your original submission. To be accepted for publication, please reconfigure your datasheet paying close attention to the reviewer's commentary. This is essential for clarity and for accessibility. There are instances in the text where stating clearly the analytic results, including specific data, will better support your argument or require rewording of your argument to a position that is fully grounded in the data. There is capacity to demonstrate greater awareness of the limitations of portable XRF and the reviewer highlights some important elements with regard to data management, though it would be advisable to pay close attention throughout the text to issues they specifically highlight and other areas where critical awareness of those limits can make your argument more convincing. I look forward to considering a revised version of the manuscript that addresses these minor points for revision. 

We look forward to receiving your revised manuscript.

Kind regards,

Barry Molloy

Academic Editor

PLOS ONE

Journal Requirements:

Additional Editor Comments:

Thank you for revising your manuscript and paying close attention to the comments by both reviewers and by the editor. This version has substantially improved as a result and in most respects is suitable for publication. However, there are a few important amendments that are required before this can be accepted for publication. These primarily relate to how you present and employ your data generated with a portable XRF. I would ask you to pay close attention to the the specific details of the reviewer comments and to the wider implication of these with regard to systematic presentation and clarity, supported by data, when discussing your results.

Reviewers' comments:

Reviewer's Responses to Questions

**Comments to the Author**

1. If the authors have adequately addressed your comments raised in a previous round of review and you feel that this manuscript is now acceptable for publication, you may indicate that here to bypass the “Comments to the Author” section, enter your conflict of interest statement in the “Confidential to Editor” section, and submit your "Accept" recommendation.

Reviewer #2: (No Response)

2. Is the manuscript technically sound, and do the data support the conclusions?

Reviewer #2: Yes

3. Has the statistical analysis been performed appropriately and rigorously? 

Reviewer #2: Yes

4. Have the authors made all data underlying the findings in their manuscript fully available?

Reviewer #2: Yes

5. Is the manuscript presented in an intelligible fashion and written in standard English?

Reviewer #2: Yes

6. Review Comments to the Author

Reviewer #2: I appreciate the changes made to the text by the authors. I believe the paper in its revised form reflects more accurately the scope and aims of the study, and highlights more substantially the value of this model to the discipline. I, however, have a few additional comments for the authors to pay attention to, which I believe will strengthen the study.

Regarding S1 and S2:

1 --- Proof-of-concept tab -> S1_Database

The Proof-of-concept tab in the S1_Database file should be curated. As of now, there is what I can best describe as a raw XRF data dump. Numbers should be stored as such (they are stored text as of now, as they were with the original submission) and only the important decimals should be included, unless the authors suggest accuracy of the pXRF to 0.00000001.

For example, the value 1.5975651858007571 should be stored as 1.6. One decimal place for a value >1 % is sufficient for pXRF data and shows understanding of the technique and its limitations.

2 --- pXRF_LOD -> S1_Database

In pXRF_LOD in S1_Dataset, please explain how you got to those LODs. From what I can understand, one CRM has been analysed: BCR32.

As an example, the authors report an LOD for Co of 0.00001. How did you get to this? This is extremely low for pXRF. This is just one example. There are more LODs that need explanation.

3 --- Ref_Material_Stats -> S1_Database

In the Ref_Material_Stats tab please show clearly the measured versus certified values (e.g. in a row just above above the mean values of the 45 measurement). As of now it is impossible to judge. Also explain your LODs for the elements not included in the CRM.

Elements not detected should not be included.

It is not clear to me how the elements to oxides conversion tables fit in this tab (which is dedicated to the certified reference material stats.

In the same tab, report data either in wt% or in ppm. Please avoid scientific formats. This will make your data much more easy to read and will add strength to your study.

4 --- Reference_Material -> S1_Database

Why include all the elements that were not detected?

5 --- S2_Table

All tables in this file need detailed captions.

In-text comments (revised manuscript):

- Lines 114-155: 1) A typical handheld pXRF has a standard spot area (ellipse in shape) of ca. 3 mm in its longest side. This is a large enough area to offer representative data for a bead. Unless the bead is made of minerals containing crystals and feldspars >3 mm. 2) Additionally, what does relative superficiality mean in the context of this sentence? Superficiality of analysis area? That would depend on the sample (cut samples can be analysed). Superficiality in terms of data quality? This takes me back to a reported LOD for CO that is way too low to be realistic. Overall, this sentence seems to be purposefully vague. Please revise showing a solid understanding of the evaluation of handheld pXRF analyses.

- Line 126: Replacing “10k beads” with “some thousands of beads” takes away credibility from the text. Just state how many beads you analysed and keep it simple. Avoid vagueness.

- Line 133: MS is (not are) unless Machine learning is meant to be plural here.

- Lines 133-138: Though not strictly wrong, this paragraph reads as rather vague (again). It would be nice to have it with more relevance to the specific study at hand.

- Line 231: How and why were <lod an="" by="" guess="" informed="" replaced="" values="">

- Line 463: A small change, nevertheless an important one: “two completely different sites” -> why not just write “two sites” and avoid unnecessary wordiness. This is an example. Similar comments apply throughout the text.

- Line 563: Do you mean the archaeology of personal adornments in general or those made by minerals such as the beads? Personal adornments can be made from a variety of materials.

Once these comments are addressed, I am happy to see this study in print.</lod>

7. PLOS authors have the option to publish the peer review history of their article (what does this mean?). If published, this will include your full peer review and any attached files.

Reviewer #2: No

---

## [Author Response · Author response to Decision Letter 1]

7 Apr 2024

We would like to acknowledge the constructive tone of all criticisms made by the editor and reviewers during the review process, as well as the commitment to improving the work as a whole. We have considered all comments and addressed most of them within the revised version. In this letter, we provide responses (in blue) to each of the raised points.

Sincerely,

Daniel Sanchez-Gomez

Corresponding author (on behalf of all co-authors),

To the academic editor,

—

Thank you for your revised manuscript and for your careful attention to the comments by the reviewers and the editor of your original submission. To be accepted for publication, please reconfigure your datasheet paying close attention to the reviewer's commentary. This is essential for clarity and for accessibility. There are instances in the text where stating clearly the analytic results, including specific data, will better support your argument or require rewording of your argument to a position that is fully grounded in the data. There is capacity to demonstrate greater awareness of the limitations of portable XRF and the reviewer highlights some important elements with regard to data management, though it would be advisable to pay close attention throughout the text to issues they specifically highlight and other areas where critical awareness of those limits can make your argument more convincing. I look forward to considering a revised version of the manuscript that addresses these minor points for revision.

R:

We have addressed all the comments made by the editor and the reviewer, which we found to be appropriate and relevant to improve the final presentation of the paper.

Specifically we have:

Modified the structure of the mentioned tabs within the dataset and table S2, with particular attention to formatting, clarity of information and captioning.

Some sentences in the text have been edited or deleted to avoid excessive wording and vague affirmations that could add unnecessary noise to the discussion.

We present the exact figures of the analyses used to train the model, avoiding vagueness.

---

R:

Done. We have not modified the reference list.

Academic Editor Comments:

Thank you for revising your manuscript and paying close attention to the comments by both reviewers and by the editor. This version has substantially improved as a result and in most respects is suitable for publication. However, there are a few important amendments that are required before this can be accepted for publication. These primarily relate to how you present and employ your data generated with a portable XRF. I would ask you to pay close attention to the specific details of the reviewer comments and to the wider implication of these with regard to systematic presentation and clarity, supported by data, when discussing your results.

Reviewers' comments:

Review Comments to the Author

Reviewer #2: I appreciate the changes made to the text by the authors. I believe the paper in its revised form reflects more accurately the scope and aims of the study, and highlights more substantially the value of this model to the discipline. I, however, have a few additional comments for the authors to pay attention to, which I believe will strengthen the study.

Regarding S1 and S2:

1 --- Proof-of-concept tab -> S1_Database

The Proof-of-concept tab in the S1_Database file should be curated. As of now, there is what I can best describe as a raw XRF data dump. Numbers should be stored as such (they are stored text as of now, as they were with the original submission) and only the important decimals should be included, unless the authors suggest accuracy of the pXRF to 0.00000001.

For example, the value 1.5975651858007571 should be stored as 1.6. One decimal place for a value >1 % is sufficient for pXRF data and shows understanding of the technique and its limitations.

R:

Done. We have formatted the data by including a tighter decimal approximation and ensuring its numerical format.

2 --- pXRF_LOD -> S1_Database

In pXRF_LOD in S1_Dataset, please explain how you got to those LODs. From what I can understand, one CRM has been analysed: BCR32.

As an example, the authors report an LOD for Co of 0.00001. How did you get to this? This is extremely low for pXRF. This is just one example. There are more LODs that need explanation.

R:

The different approaches to calculate detection limits based on reference materials represent a considerable technical challenge for routine qualitative analysis of multi-element samples and for multianalyte methods such as XRF, as multiple reference materials for each analyte may be necessary to characterise the effects of interferences. (Hyslop & White, 2008; Kadachi & Al-Eshaikh, 2012; Richard & Rousseau, 2001). Getting blank samples for each different composition in our dataset also requires a challenge that is not very achievable. Instead, as a standard procedure, we have used the instrument's LoD’ for a SiO2 matrix with the soil LE fundamental parameter configuration. LoD's are specified for each array at three sigma 99.7% confidence level. The individual LoD's improve as a function of the square root of the assay time. We have used LoD’s for a measurement time of 60 seconds. It is worth noting that since LoD’s are given in parts per million (ppm), we have calculated its transformation (which can be seen in the excel formulas inside the cells) to at% so that the missing value imputation fits the input format of the ML model. In the case of Co for instance, the LoD in the above mentioned configuration is 9 ppm for 60 seconds of measuring time, and its conversion to atomic percentage is 0,00002. We agree that these values are undoubtedly over-optimistic and not susceptible to be used for more specific quantitative analyses. In fact, as mentioned in the text, they are statistically negligible near-zero values. The reason for using these for the imputation of missing values instead of just zeros is that we wanted to be as accurate as possible by replacing missing values with a constant that would produce the least bias in the distribution according to the literature (Croghan, 2003).

On the other hand, the reference material has been used for the calibration of the apparatus in successive analysis routines, in particular for the control of the parameter values in the phosphates that make up the data set, in order to be aware of possible instrumental errors.

3 --- Ref_Material_Stats -> S1_Database

In the Ref_Material_Stats tab please show clearly the measured versus certified values (e.g. in a row just above the mean values of the 45 measurement). As of now it is impossible to judge. Also explain your LODs for the elements not included in the CRM.

Elements not detected should not be included. It is not clear to me how the elements to oxides conversion tables fit in this tab (which is dedicated to the certified reference material stats. In the same tab, report data either in wt% or in ppm. Please avoid scientific formats. This will make your data much more easy to read and will add strength to your study.

R:

We have addressed this issue by:

Changing the format of the presentation of the CRM descriptive statistics, which we hope will make them easier to read. 

Added a row in the statistics table with the certified values for easy comparison with the measurements.

Avoided scientific formats

The purpose of including the conversion of oxides to elements is because the CRM composition is reported in g/kg and we wanted to perform all the calculations within the excel sheet so that the specialised public would be aware of the reason for the values presented. Within this table are included the values of the elements in wt % and their subsequent conversion to at%.

Since the reference material is used to assess the concentrations of 9 parameters (calcium, phosphorus, carbonates, fluorine, silicon, sulphur, aluminium, magnesium and iron), we have retained all values in the calculation even though some of these elements are not detectable by the XRF instrument, however, we consider that it makes sense to include them in the calculation of the conversions, as they contain values that approximate the compositional sample to 100%. 

The LoD’s used, as mentioned above, are those of the instrument and are calculated for a SiO2 matrix in the soil LE fundamental parameter configuration. They include elements not included in the reference material.

4 --- Reference_Material -> S1_Database

Why include all the elements that were not detected?

R:

We have deleted all non-detected elements from the table

5 --- S2_Table

All tables in this file need detailed captions.

R:

We have added the following description in the supplementary information section next to the file title:

This table reports the model results for the different validation subsets and for the proof of concept. The VS1_predictions and VS2_predictions tabs inform about the predictions for the two validation subsets reserved at different stages of the model development process. VS1 corresponds to a subset reserved prior to the resampling process. VS2 contains resampled (i.e. synthetic) data. The tab proof_of_concept_predictions reports the results obtained for the use case presented in this study. They correspond to data never seen by the model and are therefore assumed to be results of the actual generalisation capacity of the model.

In-text comments (revised manuscript):

- Lines 114-155: 1) A typical handheld pXRF has a standard spot area (ellipse in shape) of ca. 3 mm in its longest side. This is a large enough area to offer representative data for a bead. Unless the bead is made of minerals containing crystals and feldspars >3 mm. 2) Additionally, what does relative superficiality mean in the context of this sentence? Superficiality of analysis area? That would depend on the sample (cut samples can be analysed). Superficiality in terms of data quality? This takes me back to a reported LOD for CO that is way too low to be realistic. Overall, this sentence seems to be purposefully vague. Please revise showing a solid understanding of the evaluation of handheld pXRF analyses.

R:

The central argument of the paragraph is that the XRF technique should be complemented with other mineralogical analyses, as fluorescence is not able to reveal the molecular composition of the beads. Otherwise, we agree that the spot area (9 mm in our device) is sufficient to provide representative information on the objects analysed in this study and that the term superficiality (which referred to the depth of penetration of the method given the variability of the samples and the impossibility of homogenising them given their archaeological nature) lends itself to ambiguous interpretations. Therefore, we consider that this sentence adds unnecessary noise to the purpose of explaining the complementary use of analytical techniques to obtain a detailed compositional and mineralogical characterisation of the objects, so we have chosen to keep the original version, which did not raise any comments.

- Line 126: Replacing “10k beads” with “some thousands of beads” takes away credibility from the text. Just state how many beads you analysed and keep it simple. Avoid vagueness.

R:

We have simplified the sentence by including strictly the data used to train the model, avoiding vague and overstated statements. We hope that this clarification meets the reviewer's expectations, while providing accurate information on the database.

The sentence has therefore been amended as follows:

Before:

Over the past decades, we have analysed some thousands of beads beads by pXRF and, when possible, by XRD, creating the largest geochemical data set (compositional and mineralogical) on personal adornments from Iberia (S1 Dataset)

After:

In recent years we have created a database that to date has (n=1243) compositional and mineralogical analyses by pXRF and XRD of elements of personal adornment from the Iberian Peninsula.

- Line 133 : MS is (not are) unless Machine learning is meant to be plural here.

R: Done

- Lines 133-138: Though not strictly wrong, this paragraph reads as rather vague (again). It would be nice to have it with more relevance to the specific study at hand.

R:

We have rephrased this sentence as follows for the purpose of greater precision:

Before: 

Machine learning (ML) are reshaping archaeology by automating existing workflows of data analysis and introducing innovative approaches that can potentially reveal macro-archaeological patterns thanks to the increasing computational power, the accessibility of algorithms in addition to the vast amounts of data that are constantly emerging and because most archaeological problems can be formalised in some way as classification or regression tasks for which algorithms are particularly well suited

After:

Machine learning has become a powerful alternative in archaeology thanks to the increasing computational power, the accessibility of algorithms and the growing amount of data that is constantly emerging.

- Line 231: How and why were

R:

To address this issue, we add the following sentence:

To calculate the imputation values, we used the device's limits of detection (LoD) for a SiO2 matrix and estimated their conversion from parts per million (ppm) to atomic percentage to fit the input format of the model (S1_Dataset). Although these are near-zero values negligible from a statistical standpoint , we wanted to replace the missing values with a constant that would produce the smallest bias in the distribution according to [30].

 - Line 463: A small change, nevertheless an important one: “two completely different sites” -> why not just write “two sites” and avoid unnecessary wordiness. This is an example. Similar comments apply throughout the text. 

R: Done. We have gone through the whole text again to address this situation, as a result of which we have edited or deleted some sentences that did not add content to the discussion.

- Line 563: Do you mean the archaeology of personal adornments in general or those made by minerals such as the beads? Personal adornments can be made from a variety of materials.

R:

Line 563 refers (in the revised version with active change control) to the characterisation of new cases that will help us to map the distribution and consumption patterns of personal mineral ornaments.We have slightly modified the phrase to emphasise its application to mineral ornaments and to avoid possible confusion. Consequently the sentence has been reworded as follows:

We propose a workflow for the interpretation of the results that is being used by our team for the characterisation of new cases that will help us to map the distribution patterns of mineral adornments in the Iberian Peninsula; i) check that the predictions at both levels match ii) check the confidence of the probability of the prediction and set a confidence threshold (we propose ∼85%) to maintain the label generated by the model.

Once these comments are addressed, I am happy to see this study in print.

References:

Croghan, C. W. (2003). Methods of Dealing with Values Below the Limit of Detection using SAS. US-EPA. https://cfpub.epa.gov/si/si_public_record_report.cfm?Lab=NERL&dirEntryId=64046

Hyslop, N. P., & Whit

---

## [Editor Report · Decision Letter 2]

9 Apr 2024

A supervised multiclass framework for mineral classification of Iberian beads

PONE-D-23-39546R2

Dear Dr. Sanchez-Gomez,

We’re pleased to inform you that your manuscript has been judged scientifically suitable for publication and will be formally accepted for publication once it meets all outstanding technical requirements.

Kind regards,

Barry Molloy

Academic Editor

PLOS ONE
---

## [Editor Report · Acceptance letter]

29 May 2024

PONE-D-23-39546R2 

PLOS ONE

Dear Dr. Sanchez-Gomez, 

I'm pleased to inform you that your manuscript has been deemed suitable for publication in PLOS ONE. Congratulations! Your manuscript is now being handed over to our production team.

Kind regards, 

on behalf of

Dr. Barry Molloy 

Academic Editor

PLOS ONE